# Lifestyle Practices and Mental Health in Adolescents: Explorative Analysis from Malaysian Health and Morbidity Survey 2017

**Irma Liyana Mushaddik [1], Karniza Khalid [1,*] , Amalina Anuar [1] , Siti Zulaiha Che Hat [1] and Ruzita Jamaluddin [1,2]**

1    Clinical Research Centre, Hospital Tuanku Fauziah, Ministry of Health Malaysia, Kangar 01000, Malaysia
2    Department of Psychiatry and Mental Health, Hospital Tuanku Fauziah, Ministry of Health Malaysia, Kangar 01000, Malaysia
*    Correspondence: karniza.khalid@moh.gov.my; Tel.: +60-4973-8000 (ext. 8413)

**Abstract:** Many significant psychosocial problems may go undetected and untreated in adolescents. We aim to determine the prevalence of high-risk behaviors among Malaysian adolescents. Retrospective data analysis was performed using data from the National Health and Morbidity Survey (NHMS) 2017 report. The dataset included 27,497 responses from adolescents aged 13 to 17 years old from five established topographic zones: northern peninsular, east coast peninsular, southern peninsular, central, and east Malaysia. The strength of associations between selected high-risk behaviors and different topographical zones were performed using logistic regression analysis. Adolescents from the northern peninsular reported the highest prevalence of suicidal attempt (27.1%) and active drug user (30.1%). The same region reported the highest prevalence of those who ever had sexual intercourse (SI) (27.2%), with 32.5% who had their first SI before 14 years old, while east Malaysia reported the highest prevalence of current smokers (26.1%) and current alcohol consumers (30.6%). Overall poor lifestyle choices were evident in the northern peninsular region. Hence, specific districts breakdown may enable targeted interventional lifestyle strategies for adolescents at risk.

**Keywords:** adolescent; prevalence; Malaysia; risk-taking; lifestyle





## 1. Introduction

Adolescence is a transition phase towards developing an adult identity that calls for intense commitments, resilience, having clear personal goals, and psychosocial well-being [1]. The phase is distinguished by rapid physical growth that coincides with puberty, as well as profound psychological changes. Adolescents could potentially be overburdened with stress due to unprecedented demands from peers, parents, school, and society [1,2]. They are reportedly self-conscious and interested in how others perceive them [1]. During this stage of vulnerability, they become more sensitive towards social evaluations and will begin to identify themselves in distinct social circles [2].

Peer relationship issues such as bullying, peer rejection, and loneliness are prevalent during the adolescence period and are known risk factors for the development of mental health issues, including depression [3,4]. In developing nations such as Malaysia, increasing prevalences of psychological distress among adolescents were evident in the last decade that were attributed to academic stress, relationship issues, and socioeconomic changes [5]. The most common high-risk behaviors among Malaysian adolescents observed were smoking (*n* = 397, 13.27%) and alcohol consumption (*n* = 133, 4.45%) [6]. A similar alarming trend was also observed in our neighboring nation, Indonesia, with up to 20% from a total of 9969 adolescents surveyed in 2015 who claimed to have been a victim of school bullying [7].

In this era of social media where online platforms are reporting robust growth, adolescents are vulnerable to intense emotional turbulence as they seek social acceptance and recognition [8]. Adolescents also have a strong tendency to engage in risky behaviors such

as smoking, alcohol consumption, illicit drug use, bullying, and having unprotected sexual intercourse due to their exploring nature and desire for novelty [9]. Unfortunately, often individuals who engage in one risky behavior are more likely to engage in others [10]. These pattern of lifestyle practices will not only have a strong influence on an individual's future personality and psychological wellbeing [11], but also may result in long-term mental health consequences.

Therefore, identifying the risk factors that contribute to poor lifestyle choices and mental health problems in adolescents is necessary to plan future preventive strategies [12]. This paper aims to determine the pattern of lifestyle practices, prevalence of high-risk behaviors, and mental health problems among Malaysian adolescents from the Malaysian Health and Morbidity Survey 2017.

## 2. Methods

### 2.1. Setting and Design

We performed retrospective data analysis using published data from the Malaysian National Health and Morbidity Survey (NHMS) 2017. Reports were available from a freely accessible public domain hosted by the Malaysian Institute for Public Health [13].

### 2.2. Data Collection

Multistage stratified cluster sampling was performed on schooling adolescents aged 13 to 17 years old from all 13 Malaysian states and 3 federal territories. A standardized questionnaire (Supplementary Materials) was distributed from 26 March 2017 to 2 May 2017. Individual and parental consents were obtained before conducting the surveys.

### 2.3. Statistical Analysis

Statistical analyses were performed using IBM SPSS Statistics version 26.0. Descriptive statistics was used to illustrate the prevalence of selected lifestyle practices among adolescents based on Malaysian topographical zones. Associations between selected high-risk behaviors and topographical zones were performed using Pearson chi-square test of independence while the strength of associations were evaluated using logistic regression analysis (expressed using crude odds ratio and 95% confidence interval).

## 3. Results

Our analysis included data from a total of 16 Malaysian states and federal territories, further divided into five Malaysian zones based on established topographic distribution: northern peninsular (Perlis, Kedah, Penang, and Perak), east coast peninsular (Kelantan, Terengganu, and Pahang), southern peninsular (Melaka, Negeri Sembilan, and Johor), central (Kuala Lumpur, Putrajaya, and Selangor), and east Malaysia (Sabah, Labuan, and Sarawak). Data included a total of 27,497 responses from adolescents aged 13–17 years old.

### 3.1. Mental Health Problems

A majority of adolescents from the central region of Malaysia reported suicidal ideation (*n* = 609, 24.6%), mostly from Kuala Lumpur (*n* = 243, 39.9%); while adolescents in the northern peninsular region had the highest prevalence of suicidal plan (*n* = 475, 25.6%) and suicidal attempt (*n* = 473, 27.1%), mostly from the state of Perak.

### 3.2. Drug Use

The northern peninsular had the highest prevalence of ever using drugs (*n* = 290, 27.5%), currently using drugs (*n* = 241, 30.1%), current use of marijuana (*n* = 179, 31.1%), and ever used methamphetamine/amphetamine (*n* = 186, 30.8%), with most being from the state of Perak. Stealing was the commonest source of drug acquisition in east coast peninsular (*n* = 34, 29.8%), while buying from the black market (*n* = 62, 30.7%) and peers (*n* = 30, 36.6%) were the commonest source of drug acquisition in the northern peninsular.

*3.3. Sexual Behaviour*

The northern peninsular reported the highest prevalence of adolescents who ever had sexual intercourse (SI) ($n$ = 520, 27.2%), had a first SI before 14 years of age ($n$ = 189, 32.5%), currently sexually active ($n$ = 374, 26.1%), ever used condom ($n$ = 68, 29.2%), and had used other types of birth control ($n$ = 62, 29.4%). A majority of these responses came from Perak.

*3.4. Violence*

The northern peninsular region had the highest incidence of adolescents reporting as violence victims ($n$ = 1559, 23.1%) and being bullied in the past 12 months ($n$ = 1,008, 22.7%). They also reported the highest incidence of being physically abused ($n$ = 726, 23.9%) and verbally abused ($n$ = 2676, 22.9%). Adolescents from the central peninsular of Malaysia reported the highest incidence of being made fun for physical appearance ($n$ = 196, 27.3%) while the highest incidence of being made fun with sexual jokes, comments, or gestures ($n$ = 130, 22.1%) and being physically abused by either being kicked, pushed, or shoved around ($n$ = 100, 27.8%) were reported by adolescents from the northern peninsular.

*3.5. Alcohol Consumption*

East Malaysia reported the highest prevalence of alcohol consumption among 12 to 13-year-olds ($n$ = 537, 34.3%), current alcohol consumer in the past 30 days ($n$ = 757, 30.6%), and adolescents reporting social issues related to their drinking problems ($n$ = 351, 29.7%). These social issues include getting into trouble with family or friends, missed school, or getting into fights, because of drinking alcohol.

*3.6. Tobacco Use*

East Malaysia reported the highest prevalence of current smokers among adolescents ($n$ = 1040, 25.4%), adolescents who tried a cigarette at a young age of less than 14 years old ($n$ = 692, 26.2%) and those who had tried an e-cigarette at the age of less than 14 years old ($n$ = 398, 27.2%). Among these, adolescents from Labuan reported the highest prevalence of ever tried a cigarette ($n$ = 255, 36.8%) and an e-cigarette ($n$ = 152, 38.2%) at the age of younger than 14 years old. Adolescents from the northern peninsular region reported the highest incidence of truancy in the last 30 days ($n$ = 2037, 25.5%), a majority of which were from Perlis ($n$ = 573, 28.1%).

*3.7. Protective Factors*

Adolescents from the northern peninsular also reported the highest prevalence of peer support ($n$ = 3036, 24.3%), parental/guardian supervision ($n$ = 925, 24.9%), connectedness ($n$ = 2229, 24.4%), and bonding ($n$ = 2924, 24.7%). The least reported prevalence of peer support ($n$ = 2240, 17.9%) and supervision from guardian ($n$ = 606, 16.3%) was from central Malaysia while east Malaysia reported the least for connectedness ($n$ = 1580, 17.3%) and bonding ($n$ = 1960, 16.6%) over the past 30 days.

We also found that the prevalence of suicidal attempt in the last 12 months, ever used drugs (not including prescribed medications), ever had sexual intercourse, current alcohol consumer in the last 30 days, and current smoker in the last 30 days were significantly associated with different Malaysian topographical zones ($p$ < 0.001) (Table 1).

**Table 1.** Associations between selected high-risk behaviors and established topographical zones in Malaysia.

| Variable(s) | Suicidal Attempt * | Ever Use Drugs * | Ever Had SI * | Current Alcohol Consumer * | Current Smoker * |
|---|---|---|---|---|---|
| | | | *n* (%) | | |
| Northern peninsular | 473 (27.1) | 290 (27.5) | 520 (27.2) | 658 (26.6) | 927 (22.7) |
| East coast peninsular | 304 (17.3) | 221 (20.9) | 384 (20.1) | 332 (13.4) | 898 (22.0) |
| Southern peninsular | 260 (14.8) | 114 (10.8) | 283 (12.4) | 381 (15.4) | 595 (14.5) |
| Central Malaysia | 338 (19.3) | 161 (15.3) | 319 (16.7) | 348 (14.1) | 630 (15.4) |
| East Malaysia | 380 (21.5) | 219 (25.5) | 408 (23.6) | 757 (30.6) | 1040 (25.4) |

* $p < 0.001$. Note: Prevalence was calculated based on the number of reported health risks by each topographical zone divided by the total number of reported health risks.

We further conducted regression analysis to determine the strength of association between the different topographical zones in Malaysia and selected high-risk behaviors (Table 2).

**Table 2.** Regression analysis to determine the strength of associations between selected high-risk behaviors and established topographical zones in Malaysia.

| | Northern Peninsular | East Coast Peninsular | Southern Peninsular | Central Malaysia | East Malaysia | *p*-Value |
|---|---|---|---|---|---|---|
| | | | Crude OR (95% CI) | | | |
| Suicidal attempt | 1.44 (1.23, 1.68) | 1.24 (1.04, 1.47) | 1.00 (Ref.) | 1.40 (1.18, 1.65) | 1.54 (1.31, 1.81) | <0.001 |
| Ever use drugs | 2.02 (1.62, 2.51) | 2.08 (1.65, 2.61) | 1.00 (Ref.) | 1.50 (1.18, 1.92) | 2.02 (1.60, 2.54) | <0.001 |
| Ever had SI | 1.46 (1.26, 1.70) | 1.45 (1.24, 1.70) | 1.00 (Ref.) | 1.20 (1.02, 1.41) | 1.52 (1.30, 1.78) | <0.001 |
| Current alcohol consumer | 1.38 (1.21, 1.57) | 0.91 (0.78, 1.06) | 1.00 (Ref.) | 0.96 (0.83, 1.12) | 2.22 (1.95, 2.53) | <0.001 |
| Current smoker | 0.80 (0.70, 0.91) | 0.99 (0.86, 1.12) | 1.00 (Ref.) | 0.90 (0.78, 1.03) | 1.47 (1.30, 1.65) | <0.001 |

Note: $p < 0.05$ is considered statistically significant.

## 4. Discussion

Our study provided important epidemiological data to reflect on adolescents' lifestyle practices in the developing Malaysian nation. We found an alarming number of adolescents who were involved in poor lifestyle practices, had significant mental health issues, early involvement with drugs, cigarettes, and alcohol, and delinquency. The Malaysian northern peninsular region recorded the highest overall prevalence for violence, early sexual activity, and drug use, central Malaysia had concerning records of mental health issues with a suicidal theme, while adolescents in east Malaysia were primarily involved with smoking and had drinking issues.

The Malaysian northern peninsular borders with Thailand, where drug trafficking and smuggling are common. The availability of substances in the region predisposes to its abuse prevalence among the locals, even garnering attention from younger audiences. Furthermore, drugs are often interlocked with promiscuity throughout human history, paving ways to chains of social ill and health risks [14]. On the other hand, central Malaysia had better infrastructure and materialistic development, shifting the mentality of its occupants to high living standards. Mid-adolescence is the most challenging time when they learn to compare self-attributes, resulting in acute stress reaction when they lack confidence when compared to others. This may render the feeling of helplessness and failure, with entailed intrapsychic and affective consequences [15]. On the other hand, smoking and drinking issues in east Malaysia had a strong influence from the cultural habits of the elderly locals [16] that also infiltrate younger generation.

The current trend in adolescents' lifestyle and practices in Malaysia are indeed alarming when compared to previous data [6,17,18]. For example, assessment of mental health problems among 5 to 15-year-old Malaysian children in 2015 found that 19.7% had internalizing issues such as peer and emotional problems [17] compared to our finding of 24–27% adolescents who reported mental health issues with a suicidal theme in 2017. The same paper also found that increasing age acted as a protective factor against mental health problems [17], which did not agree with our current analysis. These developing health risk behaviors may set a bad example for the next generation if they are not intervened early. The Malaysian health system practices a passive approach in the management of adolescents' health. Adolescents' health clinics were established in local public health facilities since 1996 under the Family Health Development Division, Ministry of Health, catering to the 10–19 years old who voluntarily seek for help [19]. Medical assessment and treatment were provided for various health-related issues in adolescents including, but not limited to, physical health (i.e., acne, migraine, and asthma), dietary issues (i.e., obesity, underweight, and anemia), mental health (i.e., stress, depression, family issues, anorexia nervosa, and bulimia), behavioral issues (i.e., smoking, alcohol, and bullying), and sexual and reproductive health (i.e., dysmenorrhea, sexually transmitted diseases, and teenage pregnancy). Despite the extensive coverage, the clinics required voluntary participation and attendance, which rendered the service underused [20].

Adolescents require innovative strategies in the promotion of mental health and preventing illness. Given that local culture has a significant influence on lifestyle decisions, targeted community interventions would be beneficial, together with the participation of society and policymakers. Since adolescents spend most of their waking hours in schools and colleges, these would be the best place to target those at risk. Identifying those who needed help may be sourced to the homeroom teacher or their class peers, through observation or local anonymous survey. A special taskforce could then be formed between the schools and relevant agencies (including medical personnel, local welfare department, and drug control authority) to ensure holistic interventional strategies, which may include but not be limited to a mental health awareness campaign to dilute social stigma and promote effective health-seeking behavior [21], rehabilitation and social camps to promote positive school environment and empower self-discipline, an individual one-to-one psychotherapy session, and family conferencing.

Our data provided corroborated evidence that adolescents' health in developing regions requires active intervention [5–7]. Missing these warning signs (i.e., early involvement with drugs, alcohol, and smoking) are associated with social ill that could have far-reaching consequences in the future.

Despite the nationwide, population-based data, our study was limited due to its methodological approach of being an observational study. The voluntary nature of the survey may hinder a full response rate and the survey may underestimate those with negative health behaviors. Therefore, we suggest for future research to embark on a more robust study design, such as an action research study to evaluate the effectiveness of a unique interventional strategy that would benefit improvement in adolescents' specific lifestyle practices and overall mental well-being. We also propose using digital data collection method via social media to increase study participation in future surveys. To the best of our knowledge, this is the first paper of its kind that provided Malaysian data by looking at specific topographical zones, hence allowing better public health strategic planning in specific localities.

## 5. Conclusions

Our study highlights the increasing prevalence of psychosocial issues and mental health among Malaysian adolescents in recent years, hence a need to focus on adolescents' health, particularly in developing nations. Multiagency involvement through clinical, public, and social health measures should begin to oversee adolescents' development

through layers of assessment to ensure they are not making poor lifestyle choices and to prepare them for vignettes of challenges ahead.

**Supplementary Materials:** The following supporting information can be downloaded at: https://www.mdpi.com/article/10.3390/adolescents2040036/s1.

**Author Contributions:** Conceptualization, I.L.M. and K.K.; Methodology, I.L.M. and S.Z.C.H.; Software, K.K.; Validation, K.K. and A.A.; Formal analysis, K.K. and I.L.M.; Investigation, A.A. and R.J.; Resources, A.A.; Data curation, I.L.M. and S.Z.C.H.; Writing—original draft, K.K., I.L.M., A.A. and S.Z.C.H.; Writing—review and editing, R.J.; Visualization, K.K., A.A. and R.J.; Supervision, R.J.; Project administration, I.L.M. All authors have read and agreed to the published version of the manuscript.

**Funding:** This research received no external funding.

**Institutional Review Board Statement:** The study was registered with the National Medical Research Register of the Ministry of Health Malaysia (NMRR ID-22-01080-ZWN). Ethical approval was waived for this study due to meeting the criteria for exemption per Malaysian National Institutes of Health (NIH) Guidelines for Conducting Research in Ministry of Health (MOH) Institutions & Facilities (3rd edition) for the use of publicly available data.

**Informed Consent Statement:** Informed consent was obtained from all subjects involved in the study.

**Data Availability Statement:** Data is available upon request.

**Acknowledgments:** The authors would like to thank the Director General of Health Malaysia for his permission to publish the paper.

**Conflicts of Interest:** The authors declare no conflict of interest.

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
