# Peer review of "Lifestyle Practices and Mental Health in Adolescents: Explorative Analysis from Malaysian Health and Morbidity Survey 2017"

_adolescents, doi:10.3390/adolescents2040036_

Round 1
Reviewer 1 Report
This manuscript reports an interesting study focusing on Malaysian adolescents' health risk behaviours based on a nationally representative survey. The use of this national data is a clear strength of the work and the large sample size, and its' representativeness is strength. I do, however, have a number of concerns about the manuscript in its present form.
General Comments.
Writing Style: parts of the manuscript are very informally worded, using a journalistic tone and various ambiguous/vague statements (e.g. "occult dele-terious social consequences on our youths"). I am unsure if the authors are native English writers, but manuscript requires rewriting to use an appropriate, objective, academic tone.
Analysis: Given the sample size, I was puzzled why no inferential statistics were conducted to explore whether some of the differences in prevalence of the health risks were conducted. It is a little difficult to follow why the analyses/descriptive statistics are meaningful for a number of reasons (a lack of clarity on whether these are statistically significant differences, what the other regions' data look like in comparison). The authors could consider presenting more of this data in table format by region so the reader can better understand where some prevalences are higher/lower.
Survey items: There is virtually no information in the manuscript (ie. the Methods) about the item wording, especially the timeframe detailed in some of the items (e.g. were some items measuring lifetime or more recent behaviours, such as the past month use of substances). There needs to be more detail in the Methods section about (1) the data and specific items that were extracted from the national survey dataset, and (2) how these items were worded so it is easier to understand the behaviours measured (e.g. lifetime vs recent use...)
Context: As a European reader, I found it difficult to understand how this data compares to other Malayasian studies (or data from surrounding countries) and what the local context is in Malaysia in terms of these health behaviours and health risks. The Introduction and Discussion sections need to cite a broader range of relevant literature; and the Introduction requires a greater discussion of the local context in Malaysia (why these are particular concerns in the country, whether Malaysia has any unique issues associated with these behaviours) which would also help strengthen the justification for the focus on Malaysia. The authors need to explain the context in the Introduction and in the Discussion make it clearer to the reader how their findings compare to similar studies from Malaysia or the surrounding region.
I would recommend that the authors consider conducting inferential statistics to understand the differences between regions/areas in terms of the data; improve the quality of the written text (use a more objective style); cite/discuss more relevant literature; present data in tables by region/area; more clearly explain the context in Malaysia in terms of these health risks and provide a clearer discussion of the findings in terms of the local context; provide a clear strengths/limitations section (e.g. acknowledging limitations... such as the data being from 2017...).
Specific Comments:
Throughout the article (including the Abstract) make sure that the unit of measurement for age is clear (i.e. years).
Abstract: the authors mention higher prevalence of some risk behaviours in Northern areas of Malaysia, but the abstract itself doesn't clarify why this is important/significant.
Introduction, first sentence: the references to adolescence being associated with intense commitments, etc., needs to be better written (what are these outcomes/changes) and there need to be references to supporting literature for these points (e.g. adolescence being associated with challenges with wellbeing)
Introduction, first paragraph: what is a 'family orientated compass'? This is an example of the vague/informal writing I mentioned earlier.
Introduction: "Poor coping skills with improper guidance, hence would result in negative responses." This sentence is incomplete and needs reference to supporting literature - there needs to be clarification what coping skills, guidance, and negative responses are being obliquely referred to here.
Introduction: as mentioned above, it would be useful to have a more developed justification for the current study (and the focus on Malaysian adolescents) in the last paragraph of the Intro. At present, it is not that clear or compelling why a focus on Malaysian adolescents is useful or important.
Method section: you mention ethics approval in the acknowledgement area of the manuscript, but not in the Methods section itself. Please add a sentence in the Methods (where appropriate) outlining the approvals obtained and any specific considerations made for your analysis from an ethics perspective.
Method/Population: There needs to be a clearer statement justifying how you divided the data into the regions mentioned - are these established regions of Malaysia or regions you devised based on the dataset?
Results (e.g. under Mental Health problems): there is a mention here for a majority of adolescents... but the cited % figure is not a clear majority. I think this might be a typo error? (c.25% does not consistute a numerical majority of the sample or subsample)
Results/top of page 3: because the manuscript does not clearly describe the survey item wording, it is not immediately clear here what 'drug' refers to (are these all illicit/illegal drugs? What was measured, in what timeframe?).
Results/violence section: what sorts of violence were 'violence victims' subject to? Again, the lack of clarity about the study/survey measures doesn't help the readers' understanding of what you are reporting.
Results/alcohol: please clarify what 'current alcohol consumer' means in terms of a measure/timeframe. Please also clarify what 'social issues' are in the context of this survey (examples of the types of issues you mean would help)
Figure 1: I found this figure to be unhelpful in terms of understanding the data/findings, I would suggest deleting this from the manuscript.
Discussion: please reword this section, paying attention to vague/informal writing like 'negative vibes' (what is a 'negative vibe'?), 'compared to yesteryears' (this is a problematic point to suggest when you don't compare your findings with any previous study/existing data).
Discussion: revise this section to include a clearer discussion of how your findings relate to the local context in Malaysia/its regions.
Discussion: towards the end of this section please include a strengths/limitatons section as standard practice.
Discussion: I found the mention that the findings might help prevent 'premature death' to be somewhat unsubstantiated by your study - please avoid making emotive/exaggurated claims about your data.
Conclusions: this needs to be reworded in a more specific academic style - a clearer sense of the findings of the study and their importance/relevance to the literature would be useful here (again, as standard practice).
Reviewer 2 Report
It would be very useful if the introduction included more detailed information about Malaysia (and any previous research), considering that the manuscript focuses on Malaysian adolescents.
Instead of "secondary" data analysis, it is better to use the term "retrospective" data analysis.
Why did you decide to analyze data from 2017 when there are more recent reports?
Please attach the validated questionnaire to the manuscript as "suppl material".
You don't have to repeat the same data in methods and results. Select one section where you will display the data.
In relation to the presented results, it would be extremely important to have a picture of Malaysia with the specified areas where you would mark the regions that dominate in certain categories. This presentation would make it easier for readers to follow the data. The modified figure 1 must be an integral part of the results and not of the discussion. It would also be useful to add all the data by region to the "suppl materials".
Given that there are reports up to 2022, the discussion should focus on trends in the last decade. The policy of the Malaysian health system must also be compared with the monitored outcomes in the domain of adolescent mental health. Please refine the discussion.
In the discussion, you must also present data from surrounding countries and regions in order to compare them with data from Malaysia. By comparing the data, it would be extremely important for readers, to identify the measures that contribute to better mental health in adolescents.
Based on the interventions implemented to improve mental health in the world, list those that would be applicable and useful for Malaysian adolescents.
Additional references of interest that you did not mention in your manuscript;
- Raaj S, Navanathan S, Tharmaselan M, Lally J. Mental disorders in Malaysia: an increase in lifetime prevalence. BJPsych Int. 2021;18(4):97-99. doi:10.1192/bji.2021.4
- Parameshvara Deva M. Malaysia mental health country profile. Int Rev Psychiatry. 2004;16(1-2):167-176. doi:10.1080/09540260310001635203
- Edman JL, Koon TY. Mental illness beliefs in Malaysia: ethnic and intergenerational comparisons. Int J Soc Psychiatry. 2000;46(2):101-109. doi:10.1177/002076400004600203
- Sahril N, Ahmad NA, Idris IB, Sooryanarayana R, Abd Razak MA. Factors Associated with Mental Health Problems among Malaysian Children: A Large Population-Based Study. Children (Basel). 2021;8(2):119. Published 2021 Feb 7. doi:10.3390/children8020119
- etc...
Please thoroughly revise the manuscript from scratch and resend it to the reviewers for reevaluation.
Round 2
Reviewer 1 Report
Thank you to the authors for submitting a substantially revised manuscript. The manuscript has been significantly improved, particularly in the quality of the written expression/use of academic style.
I have some further queries/comments following the recent revisions:
1. My main of comment is regarding the presentation and reporting of the Chi Square analysis on Page 6 - whilst it is pleasing to see my comments from the initial round of reviews have been considered, the reported Chi Square results lack clarity regarding where the differences lie in the compared regions. It is currently unclear which regions are significantly different in terms of the reported prevalences and so further follow-up tests (contrasts/comparisons) are required to partition the variance appropriately.
I would have preferred to see a more through analysis of the dataset and would disagree that such tests would not be informative 'because of the large sample size'.
Whilst the data the authors report is interesting, I think the manuscript may not have a substantial impact on the literature without more rigourous inferential analysis. The authors do not necessarily need to conduct such tests, but I would like to highlight that their findings would be more compelling had more rigorous analyses been conducted.
2. Data collection: the authors mention that the survey instrument which formed the basis of the manuscript was a 'validated' questionnaire - I'm not sure if the authors mean that this is a psychometrically validated questionnaire (which it isn't) or something else (e.g. standardised). Please clarify.
3. Discussion (Page 6, paragraph at the bottom of the page): the authors made a very brief, but very interesting, mention of there being some changes in Malaysian adolescents lifestyles/health behaviours since previous studies/surveys, but do not make it clear how the data is different (is this for a particular behaviour, for example?). Some unpacking of this point would be useful to get a sense of how adolescents' behaviours are changing in Malaysia.
--
Specific Comments
These are mostly related to further improving the written expression of the manuscript and are mostly minor by nature (although, the authors have clearly improved the written presentation of the work in the revised submission).
It would be useful to use line numbers to facilitate peer review.
Abstract: 'Northern peninsular reported...' could be better phrased as 'Adolescents from the northern peninsular....'
Intro/Paragraph 2: comma needed after 'Indonesia'
Intro/Paragraph 3: 'They' in the second sentence is a little ambiguous - 'Adolescents also...' would read more specifically
'Violence' section: 'Adolescents from the central peninsula of Malaysia' would read better in the third sentence
Limitations section: 'we may not pick up those who were truly problematic' could be better phrased - e.g. "the survey may not have identified those with more problematic negative health behaviours..."(?)
Final sentence, limitations section: I think the authors mean that there would be improvements to 'public health' strategic planning here?
Reviewer 2 Report
Table 1 is unnecessary considering that all the questions from the survey questionnaire are in the supplementary materials. Please delete it.
In the methods section, the "Ethical clearance" section is unnecessary, given that the same is stated at the end of the manuscript. Please delete it.
I don't see any of the suggested references. For better quality work, please include the suggested ones in your manuscript.
Form the "Authors contribution" and "References" sections in accordance with the instructions for authors.
Round 3
Reviewer 2 Report
Please make table 2 formatted in the style of table 1.
Also under table 2, write for which p is a statistically significant result.
